# Spatial and Temporal Distribution Characteristics and Potential Sources of Microplastic Pollution in China's Freshwater Environments

Hualong He [1], Sulin Cai [1], Siyuan Chen [1], Qiang Li [1], Pengwei Wan [1], Rumeng Ye [1], Xiaoyi Zeng [1], Bei Yao [1], Yanli Ji [1], Tingting Cao [1], Yunchao Luo [3], Han Jiang [1], Run Liu [1], Qi Chen [1], You Fang [1], Lu Pang [1], Yunru Chen [1], Weihua He [1], Yueting Pan [1], Gaozhong Pu [4], Jiaqin Zeng [5] and Xingjun Tian [1,2,*]

[1]   School of Life Sciences, Nanjing University, Nanjing 210023, China; hehua@smail.nju.edu.cn (H.H.); mg1930001@smail.nju.edu.cn (S.C.); mg20300060@smail.nju.edu.cn (S.C.); waterliqiang@163.com (Q.L.); w18362983051@163.com (P.W.); yrm_666@163.com (R.Y.); xyzeng@connect.hku.hk (X.Z.); by-beiyao@hotmail.com (B.Y.); luckyjiyanli@163.com (Y.J.); caotting@zafu.edu.cn (T.C.); JH269962460@163.com (H.J.); recho24@126.com (R.L.); chen1455526947@163.com (Q.C.); youfang@smail.nju.edu.cn (Y.F.); panglugxs@163.com (L.P.); cherry960502@163.com (Y.C.); 18351870527@163.com (W.H.); njupyt@163.com (Y.P.)

[2]   Co-Innovation Center for Sustainable Forestry in Southern China, Nanjing Forestry University, Nanjing 210037, China

[3]   School of Life Sciences, Shanxi Normal University, Taiyuan 030031, China; luoyunchao@sxnu.edu.cn

[4]   Guangxi Key Laboratory of Plant Conservation and Restoration Ecology in Karst Terrain, Guangxi Institute of Botany, Guangxi Zhuang Autonomous Region and Chinese Academy of Sciences, Guilin 541006, China; pukouchy@163.com

[5]   College of Resources and Environment, Tibet Agricultural & Animal Husbandry University, Linzhi 860000, China; zengjiaqin2009@163.com

*   Correspondence: tianxj@nju.edu.cn

**Abstract:** Microplastic pollution is a research hotspot around the world. This study investigated the characteristics of microplastic pollution in the freshwater environments of 21 major cities across China. Through indoor and outdoor experimental analysis, we have identified the spatial and temporal distribution characteristics of microplastic pollution in China's freshwater environments. Our findings indicate that the average concentration of microplastics in China's freshwater environments is 3502.6 n/m$^3$. The majority of these microplastics are fibrous (42.5%), predominantly smaller than 3 mm (28.1%), and mostly colored (64.7%). The primary chemical components of these microplastics are polyethylene (PE, 33.6%), polyvinyl chloride (PVC, 21.5%), polypropylene (PP, 16.8%), and polystyrene (PS, 15.6%). The abundance of microplastics in China's freshwater environments generally tends to increase from west to east and from south to north, with the lowest concentration found in Xining, Qinghai (1737.5 n/m$^3$), and the highest in Jiamusi, Heilongjiang (5650.0 n/m$^3$). The distribution characteristics of microplastics are directly related to land use types, primarily concentrated in areas of intense human activity, including agricultural, transport, and urban land. Seasonal changes affect the abundance of microplastics, peaking in summer, followed by spring and autumn, mainly due to variations in rainfall, showing a positive correlation.

**Keywords:** China's freshwater environment; microplastics pollution; spatial and temporal distribution characteristics; land use types; rainfall

## 1. Introduction

Globally, at least 300 million tons of plastic are produced annually [1], the majority of which enter the environment and remain for decades [2], posing severe risks to biological safety [3]. China, as a populous and major agricultural country [4], is also one of the largest plastic producers [5]. In 2018, China's plastic production reached 60.4215 million tons, accounting for 29% of the global total [6]. At the same time, China is one of the largest

consumers of plastic [7], generating massive amounts of plastic waste yearly [8]. The extensive use of plastic bags, fast food containers, plastic greenhouses, and agricultural film has led to a significant "white pollution" problem [9–12].

Plastics in the environment continuously degrade into microplastics (MPs) and nanoplastics (NPs) [13]. Microplastics are generally defined as plastics smaller than 5 mm [14]. The issue of microplastic pollution has been widely reported worldwide and has become a significant environmental problem, attracting public attention [15,16]. Due to their ubiquity, persistence, and potential ecological risks, microplastics have become a hotspot in new pollutant research [17,18]. MPs are difficult to degrade, can adsorb other pollutants, and accumulate in the food chain, thus posing substantial hazards [19–21]. Studies have shown that microplastics can alter the structure and function of ecosystems, ultimately affecting biodiversity [22–26].

Freshwater ecosystems include large bodies of water such as rivers and lakes, as well as smaller bodies like ditches and ponds [27–29]. These ecosystems, closely related to human life, boast high species diversity and provide numerous ecological services, making them highly susceptible to microplastic pollution [30,31]. Microplastics in trash, sludge, and wastewater can enter freshwater environments directly, while those in soil may be carried into water bodies by runoff [32–34]. Although current research on microplastic pollution mainly focuses on marine ecosystems, reports have identified terrestrial ecosystems as major sources of pollution [35,36]. Freshwater environments often have a more immediate physical proximity to human activities compared to marine environments, leading to a more direct impact on water quality and human health through recreational and consumption routes [37,38]. As freshwater environments are crucial pathways for microplastics to transfer from land to sea, research on microplastics in freshwater environments is growing and has yielded important insights. However, it remains less extensive compared to studies focused on marine environments [39,40].

The study of microplastic pollution in freshwater ecosystems focuses on water bodies, sediments, and biota, with factors affecting the distribution, accumulation, and migration of microplastics being key research topics [27,30,31]. It has been demonstrated that microplastics are widely distributed in China's freshwater environments [41]. An investigation in the Yangtze River Delta of China found the widespread presence of microplastics [42]. Even in the high-altitude area on the Tibetan Plateau, microplastics are also commonly present in freshwater [43]. However, due to the diversity of China's freshwater environments, the abundance of inhabiting species, varied geographical locations, and complex socioeconomic backgrounds, data are currently widely distributed and lack systematic integrity [44]. Moreover, China's rapid economic growth and the significant contribution of plastic waste highlight the urgent need for effective policies and actions to address this issue [45]. Hence, there is an urgent need for research on microplastic pollution in China's freshwater environments [46].

This study selects 21 major cities across China, primarily located in the Yangtze and Yellow River basins, encompassing most types of land use and representing China's main freshwater environments. By investigating and analyzing the characteristics of microplastic pollution in these cities' freshwater environments, this study aims to answer the following scientific questions: (1) What are the spatial and temporal distribution characteristics of microplastic pollution in China's freshwater environments? (2) What are the potential sources of microplastic pollution in China's freshwater environments? (3) What are the driving factors affecting the distribution of microplastic pollution in China's freshwater environments?

## 2. Materials and Methods

### 2.1. Sampling Locations

Our study systematically selected 21 cities across China, focusing on both urban and suburban areas. The sampling covered both large water bodies, such as rivers and lakes, and smaller ones like ditches and ponds, encompassing naturally occurring and man-made sites. These cities were strategically chosen to represent a broad spectrum of land use types,

including but not limited to agricultural, urban, and suburban areas. This diverse selection aims to capture the varying degrees of microplastic pollution influenced by different human activities across geographical locations.

Sampling was conducted four times in 2020, during April, June, August, and October, with three replicates at each site, resulting in a total of 252 water samples. We conducted our sampling during two distinct seasons: the dry season (spring and autumn) and the rainy season (summer). This approach allowed us to capture seasonal variations in microplastic abundance [47]. Detailed information and locations of the sampling sites are provided in Figure 1 and in Tables S1 and S2.

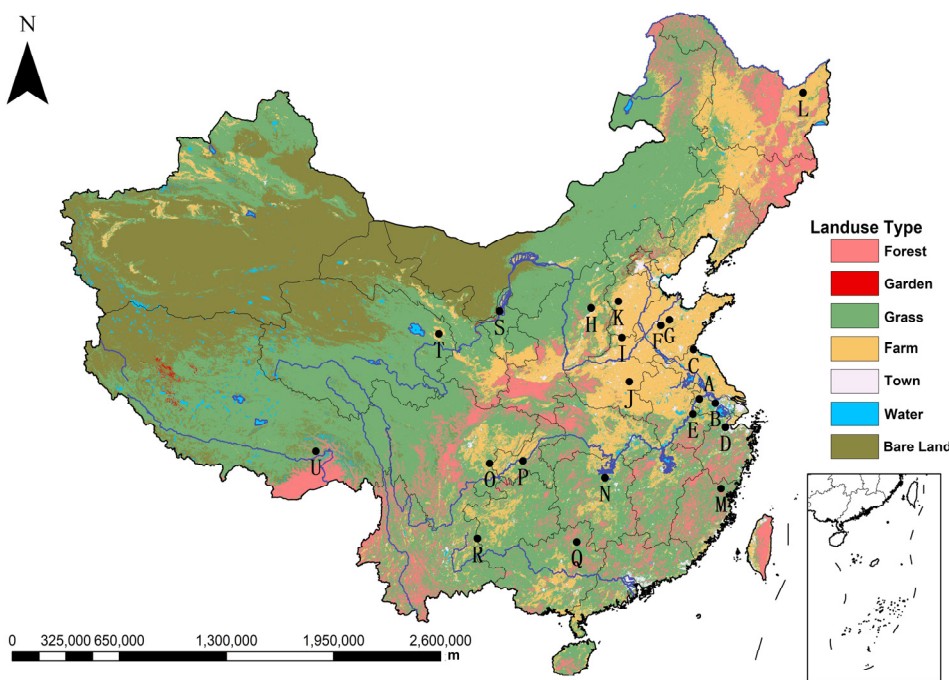

**Figure 1.** Spatial distribution of sampling locations across China. Each black dot represents a sampling site.

Winter was excluded from the seasonal sampling due to the freezing of rivers in northern China, which poses significant challenges for sampling. Additionally, the reduced biological and human activity during this season could potentially skew the representation of microplastic pollution levels.

*2.2. Sampling Method*

At each sampling site, GPS was used to determine the coordinates, and the surrounding environment and water body types were photographed for future reference. Surface water (5–10 cm deep) was collected randomly in 1000 mL glass bottles after rinsing the container with site water [48]. Each site had three replicates. The samples were then transported to the laboratory, stored at 2 °C in the dark, and sealed [49].

*2.3. Extraction of Microplastics from Water Samples*

Currently, there is a complete set of mature methodologies for the extraction and analysis of microplastics in water bodies [50–52]. Our experiment mainly referenced the work of Su and Hu [48,49], as shown in Figure S1. The extraction of microplastics from water samples involves three steps: (1) measure and record the volume of each site and its parallel water samples. (2) Using a vacuum pump (ME1, Vacuubrand, Wertheim, Germany) and a glass filter (XX1004700, Millipore, Boston, MA, USA), firstly filter the water sample onto a nylon membrane filter (NY2004700, Millipore, USA) with a diameter of 47 mm and a pore size of 20 µm, then rinse the substances on the filter membrane

into a 250 mL conical flask with 30% (*v*/*v*) hydrogen peroxide (H1009, Sigma-Aldrich, Saint Louis, MO, USA), and finally place it in a high-temperature shaker for digestion (80 rpm, 65 °C, not exceeding 72 h) until the organic matter is completely digested and the solution is transparent. (3) Filter the digested solution onto a nitrocellulose membrane filter (HAWP04700, Millipore, USA) with a diameter of 47 mm and a pore size of 0.45 μm, place it in a 6 cm diameter glass Petri dish, dry it in an oven, and seal and store it for subsequent analysis. Due to the high amounts of impurities, to ensure the accuracy of the experiment, each water sample needs to be filtered onto three membrane filters, resulting in a total of 756 nitrocellulose membrane filters.

### 2.4. Microscopic Examination and Statistical Analysis of Microplastic Samples

Microplastic samples were examined and photographed under a stereomicroscope (E100, Nikon, Tokyo, Japan) at 30–40× magnification, adjusting as needed for microplastic size. The ImageJ software (Version 1.8.0) was used for measuring and statistically analyzing the microplastic sizes. Photographs were used to classify microplastics by shape, size, color, and abundance.

### 2.5. Chemical Composition Identification of Microplastic Samples

In this experiment, micro-Fourier Transform Infrared Spectroscopy (μ-FTIR, Spectrum Two, PerkinElmer, Waltham, MA, USA) was employed to identify the chemical composition of microplastics, while a Field Emission Scanning Electron Microscope (FE-SEM, Mira 4, Tescan, Brno, Czech Republic) was used to photograph the surface structure of the microplastics. Initially, a small number of samples were selected for identification to gain experience, followed by the random selection of a large number of samples for formal identification.

The μ-FTIR identification process involved transferring samples to clean nitrocellulose membranes, placing them on the μ-FTIR sample platform, and using the OPUS software (Version 8.8) for analysis. The FE-SEM process included preparing the samples on conductive tape, coating them with gold to prevent charging, and photographing them at varying magnifications based on the sample type.

### 2.6. Data Acquisition on Population, Economy, Rainfall, and Land Use Types

The population, economic, and rainfall data used in this study were all sourced from statistical yearbooks published by various city statistics bureaus, while the data on land use types were derived from the results of China's third national land survey released by the Ministry of Natural Resources [53]. The third national land survey of China commenced in October 2017 and was completed in 2020. The survey comprehensively utilized satellite remote sensing images with a resolution better than 1 m to create base maps for the investigation. The survey, lasting three years, involved 219,000 survey personnel and compiled 295 million survey plot data points, thoroughly clarifying the status of land use in China. Therefore, the related data used in this study are reliable and credible.

### 2.7. Quality Assurance

The presence of procedural contamination or air blank contamination can impact the final results. Therefore, to minimize sample contamination during field sampling, the following measures were taken: First, before sampling, all containers and tools were cleaned with filtered water (distilled water filtered through a 47 mm diameter, 5 μm pore size filter membrane) and covered or wrapped in foil to prevent contamination. Second, gloves were worn during the sampling process. Third, samples were sealed and stored immediately after collection to avoid direct exposure to the atmosphere as much as possible.

In the laboratory analysis process, all solutions were filtered and prepared using a 47 mm diameter, 5 μm pore size polycarbonate filter membrane (TMTP04700, Millipore, USA). During the drying step, five open Petri dishes were used to estimate air blank contamination within the oven, and after being placed for 72 h, no microplastic blank

contamination was found in any of the Petri dishes, proving that the oven environment was clean. To reduce risk, samples were still slightly covered with aluminum foil during the drying process.

Additionally, we set up five blank controls to rigorously detect any system or experimental errors. These controls were tested alongside the experimental samples in identical conditions to ensure no contamination influenced our results. The analysis of these blank controls showed no detectable contamination, affirming the reliability of our experimental procedures and data.

### 2.8. Data Analysis

Data analysis and graphing were conducted using RStudio software (Version 4.3.2). The data were initially subjected to a normal distribution test (Shapiro–Wilk Test) and a homogeneity of variances test (Bartlett's Test). If the data were normally distributed, an ANOVA was used for intra-group variance analysis, followed by a Tukey Test for inter-group multiple comparisons. If the data did not follow a normal distribution, a non-parametric testing method (Kruskal–Wallis Test) was employed for intra-group variance analysis, with Dunn's Test and the Bonferroni correction method applied for inter-group multiple comparisons.

The map of vegetation in China was created using ArcGIS (Version 10.2). The map showing the distribution of sample abundance at sampling sites was generated using R extension packages (sf, ggspatial), with map vector data sourced from DataV.GeoAtlas [54]. The other figures were produced using RStudio software (Version 4.3.2) and the corresponding extension packages (tidyverse, etc.).

## 3. Results

### 3.1. Physical and Chemical Properties of Microplastic Samples

By analyzing 256 water samples through filtration and digestion, a total of 756 filter membranes were obtained. Microplastic samples on all filter membranes were observed, photographed, scanned, and identified, with results recorded (Figure 2). Under a stereomicroscope set at various magnifications, microplastics of different shapes were observed: fibers, films, fragments, and pellets. Field emission scanning electron microscopy (FE-SEM) revealed that the surfaces of these differently shaped microplastics were not smooth, with most being very rough. The chemical components of the extracted microplastic samples, identified using micro-Fourier transform infrared spectroscopy (μ-FTIR), were primarily polyethylene (PE), polyvinyl chloride (PVC), polypropylene (PP), and polystyrene (PS).

Statistical analysis of all samples yielded distribution characteristics of microplastics in terms of shape, size, and color (Table 1 and Figure 3). Table 1 presents the analysis results of all samples. In terms of shape, fibers were the most common, accounting for 42.5% of microplastics, followed by films at 38.2%. Regarding size, microplastics smaller than 0.3 mm predominated, comprising 28.1% of the total, with those smaller than 0.5 mm making up 51.9%. In terms of color, transparent microplastics were most common at 35.3%, but colored samples accounted for 64.7%, mainly white and black, indicating that most microplastics were colored.

**Table 1.** Distribution characteristics of microplastics by shape, size, and color (%).

| Shape | Percentage | Size (mm) | Percentage | Color | Percentage |
|---|---|---|---|---|---|
| Fiber | 42.5 | <0.3 | 28.1 | Transparent | 35.3 |
| Film | 38.2 | 0.3–0.5 | 23.7 | White | 20.8 |
| Fragment | 11.5 | 0.5–1 | 17.1 | Black | 13.5 |
| Pellet | 7.8 | 1–2 | 13.8 | Blue | 11.4 |
| | | 2–3 | 10.9 | Green | 9.3 |
| | | 3–5 | 6.4 | Red | 6.1 |
| | | | | Others | 3.5 |

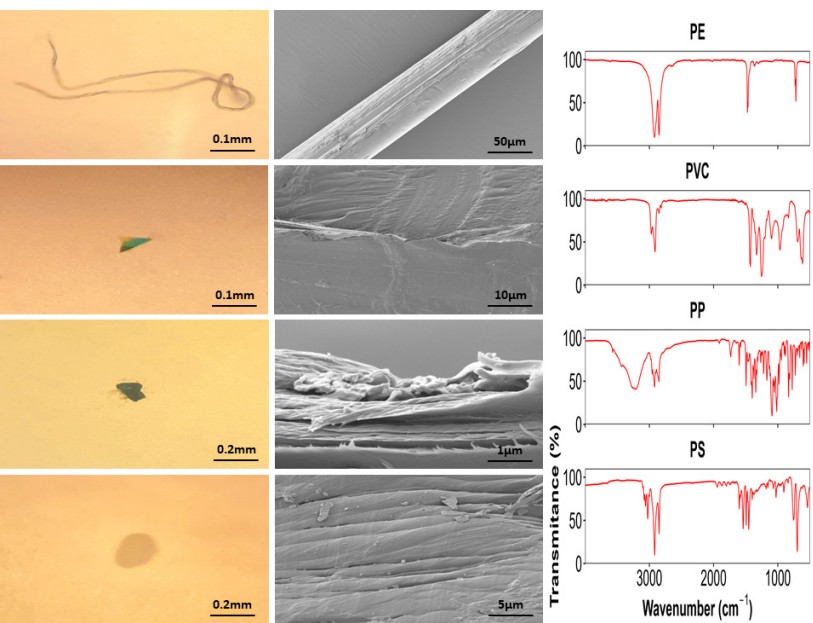

**Figure 2.** Stereomicroscope, SEM, and spectral images of microplastic samples.

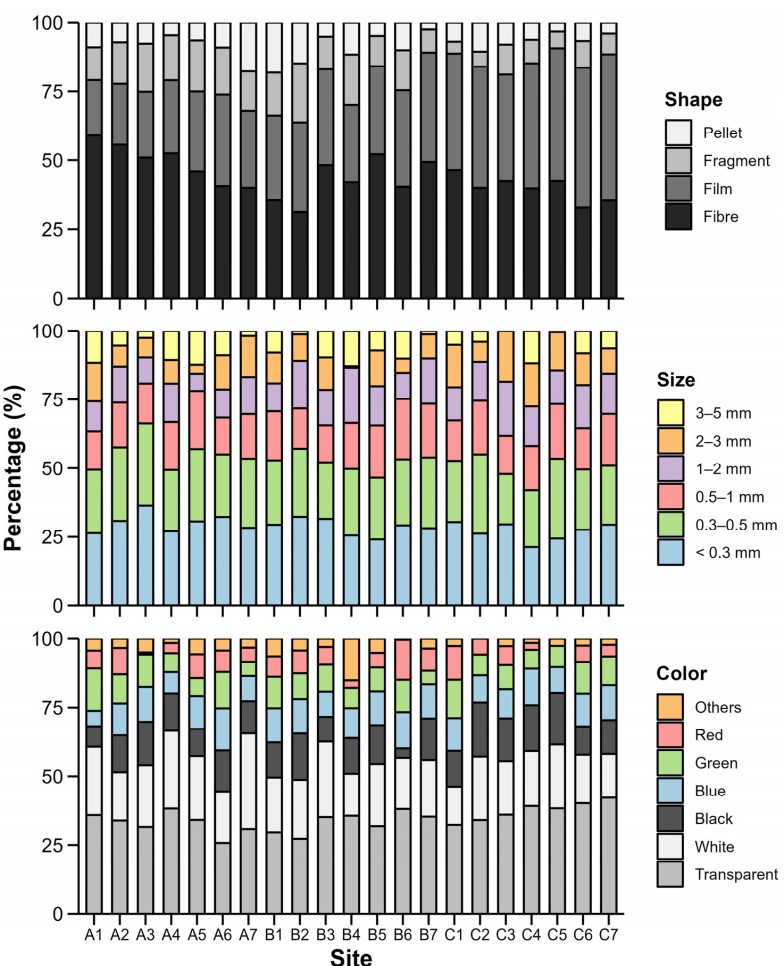

**Figure 3.** Distribution characteristics of microplastics by shape, size, and color. Group information details can be found in Table S3. All sampling sites were arranged in ascending order of microplastic abundance and divided into three groups.

The analysis of various sampling points (Figure 3) showed that in terms of shape, Group A had the highest proportion of fibers at 48.2%, surpassing the other groups. However, Group C had the highest proportion of films at 46.6%. In terms of size, Group A had the highest proportion of samples smaller than 0.5 mm at 55.4%, while Group C had a higher proportion of samples larger than 0.5 mm, indicating a larger average size. The distribution of colors among the three groups was relatively even, with no significant differences ($p > 0.05$).

Micro-Fourier transform infrared spectroscopy (μ-FTIR) was used for infrared spectroscopic analysis of all samples. The statistical analysis of these results provided the distribution characteristics of the chemical components of microplastics (Figure 4). It was found that the chemical components of the microplastic samples were mainly polyethylene (PE), polyvinyl chloride (PVC), polypropylene (PP), and polystyrene (PS), with PE being the most prevalent at 33.6%, followed by PVC at 21.5%. Figure 4 also reveals that, among all analyzed samples, there were some non-plastic components, although they only constituted 7.5% of the materials. This indicates that the experimental process extracted not only plastic samples but also some non-plastic components. While there were minor errors, they were negligible and did not impact the experimental results.

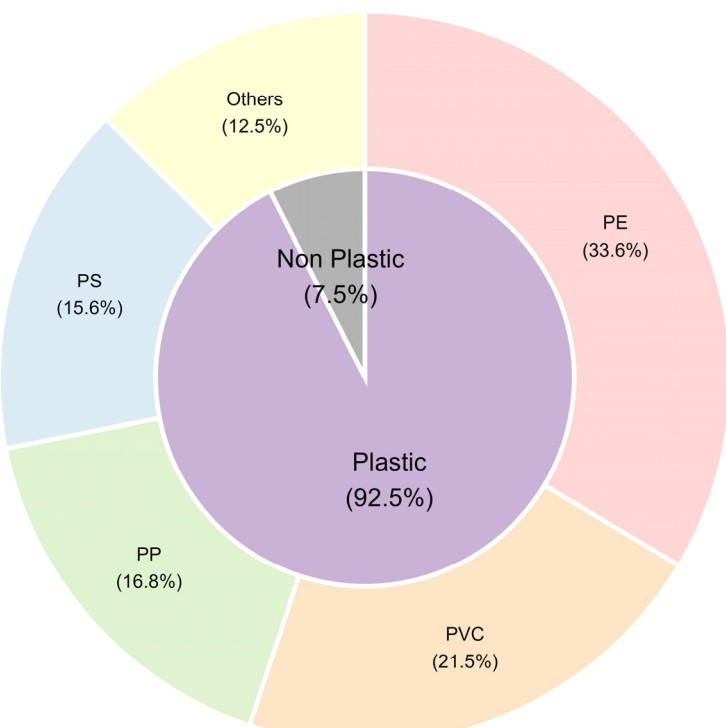

**Figure 4.** Distribution characteristics of microplastics by chemical composition. The external circular chart displays the percentage composition of different types of microplastics identified. The internal pie chart illustrates the proportion of non-plastic components detected in the samples.

### 3.2. Spatial Distribution Characteristics of Microplastic Abundance

Statistical analysis of all sampling results revealed the spatial distribution characteristics of microplastic abundance in China's freshwater environments, with an average abundance of 3502.6 n/m$^3$ (Figure 5). The lowest abundance was in Xining, Qinghai (1737.5 n/m$^3$), and the highest in Jiamusi, Heilongjiang (5650.0 n/m$^3$). Microplastics were primarily found in areas with frequent human activity, including economically developed and agriculturally intensive regions.

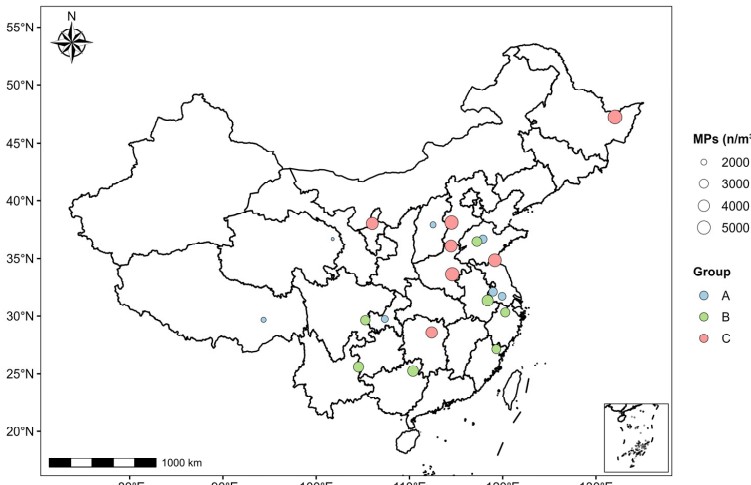

**Figure 5.** Spatial distribution characteristics of microplastic abundance. The points in the figure represent different sampling sites, with different colors indicating different groups. The varying sizes of the points denote different abundances; the larger the area, the higher the abundance.

We categorized all the sample sites into three groups according to the abundance of microplastics, ranging from lowest to highest (Table S3). Group A's sampling locations are predominantly situated in the eastern and western regions, exhibiting the lowest mean abundance of microplastics at merely 2228.9 n/m$^3$. The sampling sites of Group B are chiefly located in the southern region, demonstrating a relatively higher mean abundance of microplastics, calculated at 3346.7 n/m$^3$. Conversely, Group C's sampling locations are mainly concentrated in the central and northern regions, with the highest mean abundance of microplastics, recorded at 4932.3 n/m$^3$. Further analytical examination revealed that the differences between the groups are markedly significant ($p < 0.001$).

Subsequent regression analysis was conducted to assess the relationship between the abundance of microplastics and geographic coordinates (Figure 6). The analysis revealed that there is a general upward trend in microplastic abundance with increasing longitude, evidencing a notable positive correlation ($p < 0.05$). Similarly, an overall upward trend in microplastic abundance was observed with increasing latitude; however, the positive correlation in this case was less pronounced ($p > 0.05$). Thus, these findings allow us to deduce that the abundance of microplastics in China's freshwater environments exhibits a general increasing trend from west to east and from south to north.

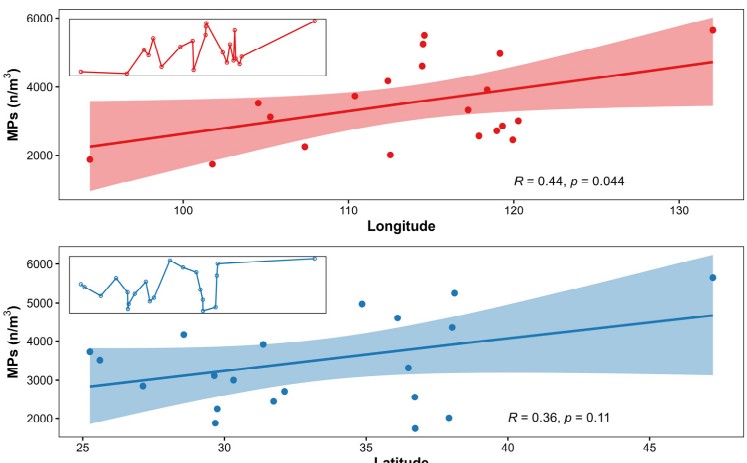

**Figure 6.** Geographic distribution characteristics of microplastic abundance. The major part of the figure is the regression analysis, with a small chart in the left corner showing the trend of microplastic abundance changes.

### 3.3. Temporal Distribution Characteristics of Microplastic Abundance

Statistical analyses of data from four sampling events elucidated the temporal distribution patterns of microplastic abundance in freshwater environments across China (Figure 7). Overall, the abundance of microplastics demonstrated an initial increase followed by a decrease within the period from April to October. Among these sampling points, the month of June recorded the peak abundance of microplastics, reaching 4776.2 n/m$^3$. This was succeeded by August, with an abundance of 3904.8 n/m$^3$, and subsequently October, showing 2771.4 n/m$^3$. The lowest abundance was observed in April, with a mere 2557.1 n/m$^3$.

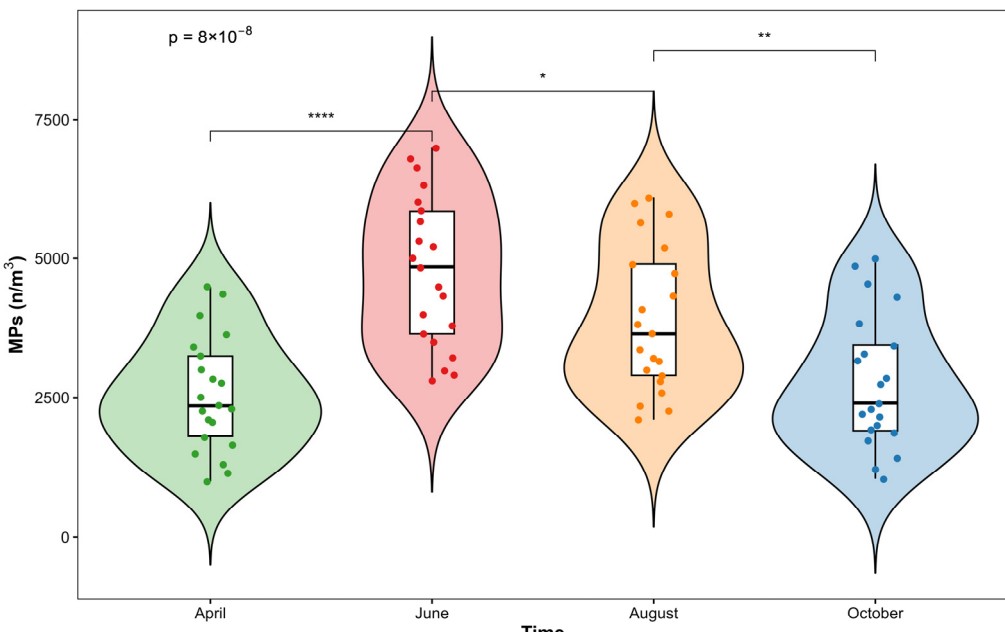

**Figure 7.** Temporal distribution characteristics of microplastic abundance. The *p*-value in the top left corner represents the difference among all groups. The number of asterisks * above the lines connecting groups indicates the size of the *p*-value; * for *p* < 0.05, ** for *p* < 0.01, **** for *p* < 0.001.

When comparing the microplastic abundances across the four sampling instances, the variations were found to be highly significant (*p* < 0.001). Specifically, the comparative analysis between adjacent sampling periods highlighted that the difference between April and June was the most pronounced, followed by a discernibly significant difference between August and October, while the gap between June and August was relatively minor.

In China, April typically corresponds to the spring season, and October to the autumn season, with both June and August falling within the summer period. Conducting an analysis based on these seasonal distinctions, the data reveal a seasonal distribution trend for microplastic abundance in China's freshwater environments: the highest abundance is recorded during the summer months, averaging 4340.5 n/m$^3$, followed by autumn with 2771.4 n/m$^3$, and the lowest in spring at 2557.1 n/m$^3$.

### 3.4. Relationship between Microplastic Distribution and Land Use Types

Figure 5 has already shown that microplastics are mainly distributed in areas with frequent human activities, including economically developed and agriculturally concentrated regions. However, the specific relationship between microplastic distribution and various types of land use, and the extent of this relationship, has yet to be determined. This section will further explore the relationship between the characteristics of microplastic distribution and land use types.

Using the R extension packages (linkET and vegan), a Mantel test was conducted to examine the relationship between the abundance of microplastics and types of land use (Figure 8). The Mantel test, a statistical method used to assess the correlation between

two distance matrices, helps us understand the relationship between the geographical distribution of microplastics and various land use types. This non-parametric test is particularly useful in ecological studies where data may not meet the assumptions required by more traditional parametric tests.

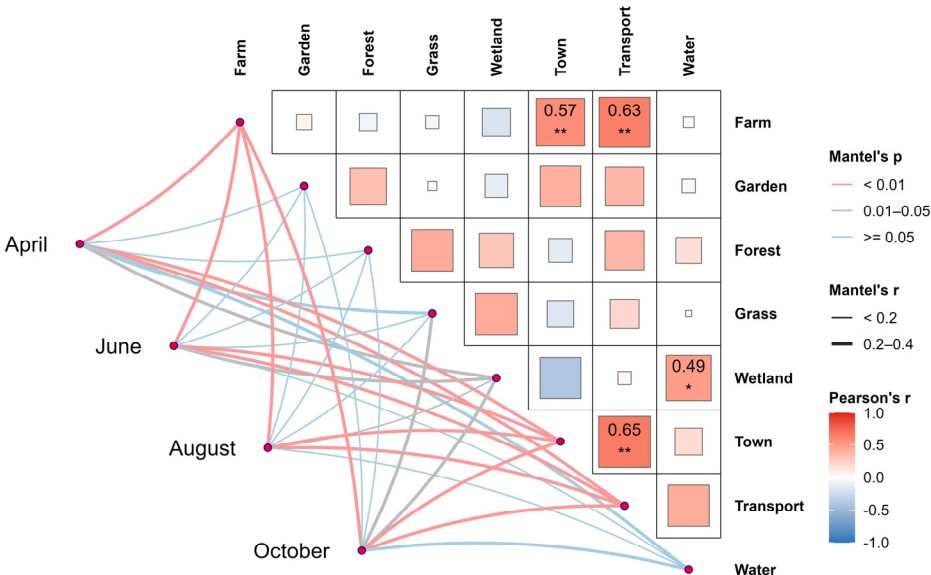

**Figure 8.** Mantel test. The lower left corner of the figure displays a correlation network diagram between microplastic abundance and land use types, with red lines indicating positive correlations and blue lines indicating negative correlations. The thicker the line, the stronger the correlation. The upper right corner shows a heatmap of correlations between land use types, with red indicating positive correlations and blue indicating negative correlations. The larger the square area, the larger the correlation coefficient. Asterisks indicate the level of significance of the correlation, * for $p < 0.05$, ** for $p < 0.01$.

The results showed a correlation between different types of land use. There was a relatively high correlation between urban and transport land ($r = 0.65$) and a considerable correlation between agricultural land and both urban ($r = 0.57$) and transport land ($r = 0.63$).

Examining the relationship between microplastic abundance and types of land use, with April sampling as an example, the highest correlation was found with agricultural land, followed by transport and then urban land. The correlation $p$-values were all less than 0.01, indicating a highly significant relationship. A weak correlation existed between microplastic abundance and wetlands, with a $p$-value less than 0.05 but greater than 0.01. No correlation was found between microplastic abundance and gardens, forests, grasslands, and water area lands, with all $p$-values greater than 0.05. Similar correlation patterns were observed in the analysis of samples from other months.

The Mantel test indicated that the distribution characteristics of microplastics are directly related to types of land use, mainly concentrated in areas with frequent human activities, including agricultural, transport, and urban land.

Further analysis through redundancy analysis (RDA) using R extension packages (ggpubr, ggrepel, and vegan) delved into the impact of land use types on microplastic distribution characteristics. The data were first subjected to Hellinger transformation for dimensionality reduction and normalization to enhance reliability.

Results (Figure 9) confirmed that agricultural, transport, and urban land have the most significant impact on microplastic abundance. This is in complete agreement with the Mantel test results, further proving that microplastic distribution characteristics are directly related to types of land use, primarily concentrating in areas of frequent human activity such as agricultural, transport, and urban land.

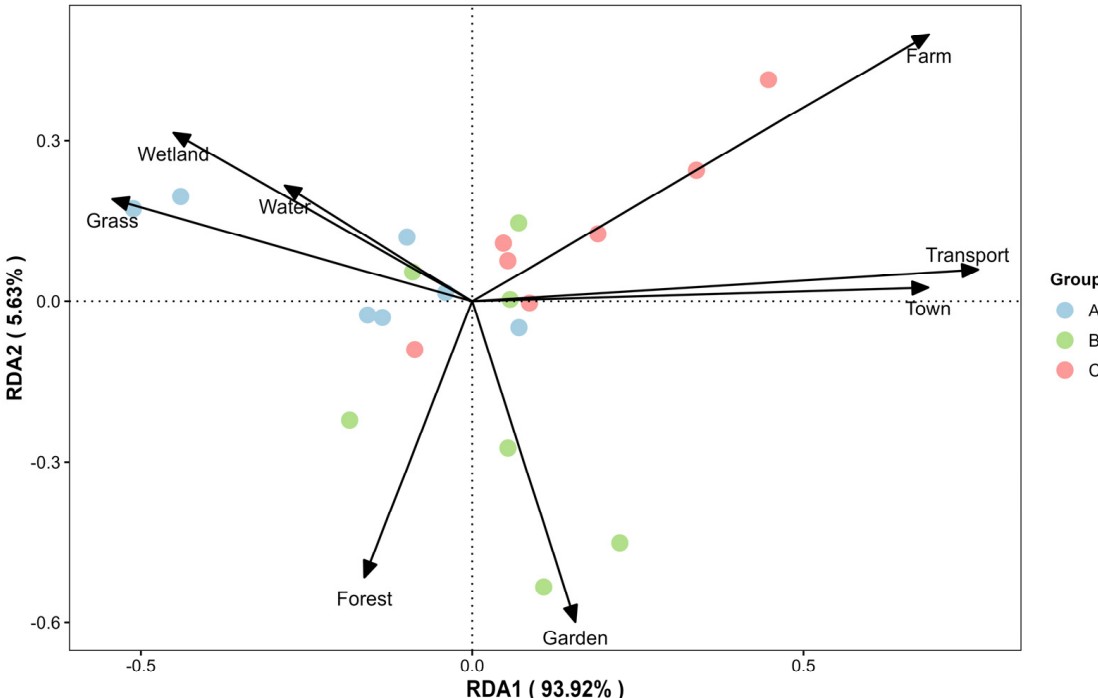

**Figure 9.** Redundancy analysis (RDA). The *x*-axis (RDA1) and *y*-axis (RDA2) represent the first and second principal components, respectively, explaining the highest proportions of variance in the data set. The points in the figure represent samples, with different colors indicating different groups, while arrows originating from the origin represent land use types. The length of an arrow indicates the strength of the impact of land use type on microplastic abundance, with longer arrows indicating a stronger influence of that land use type. The angle between an arrow and the axes represents the correlation between the land use type and the axes, with smaller angles indicating higher correlations. The vertical distance from a sample point to an arrow and its extension line indicates the strength of the impact of land use type on microplastic abundance; the closer a sample point is to an arrow, the stronger the influence of that land use type on microplastic abundance. If a sample point is in the same direction as an arrow, it indicates a positive correlation between microplastic abundance and that land use type. If a sample point is in the opposite direction of an arrow, it indicates a negative correlation between microplastic abundance and that land use type.

### 3.5. Relationship between Microplastic Distribution and Social and Natural Factors

Figure 7 has shown that the abundance of microplastics changes over time, with June showing higher abundance than other months. To understand why this variation occurs, a regression analysis was conducted between microplastic abundance and social and natural factors (Figure 10).

The results showed a negative correlation between microplastic abundance and regional GDP, indicating that economically developed areas do not necessarily have higher microplastic abundance (Figure 10A). A positive but not significant correlation ($p > 0.05$) was found between microplastic abundance and population size, suggesting that microplastic abundance tends to increase with population growth (Figure 10B).

A strong positive correlation (*p*-value close to 0.01) was found between microplastic abundance and regional area, indicating that microplastic abundance increases with the size of the area (Figure 10C). Correlation analysis between microplastic abundance at different sampling sites across various months and rainfall showed a very strong positive correlation ($p < 0.001$), identifying rainfall as the most critical factor affecting microplastic distribution (Figure 10D).

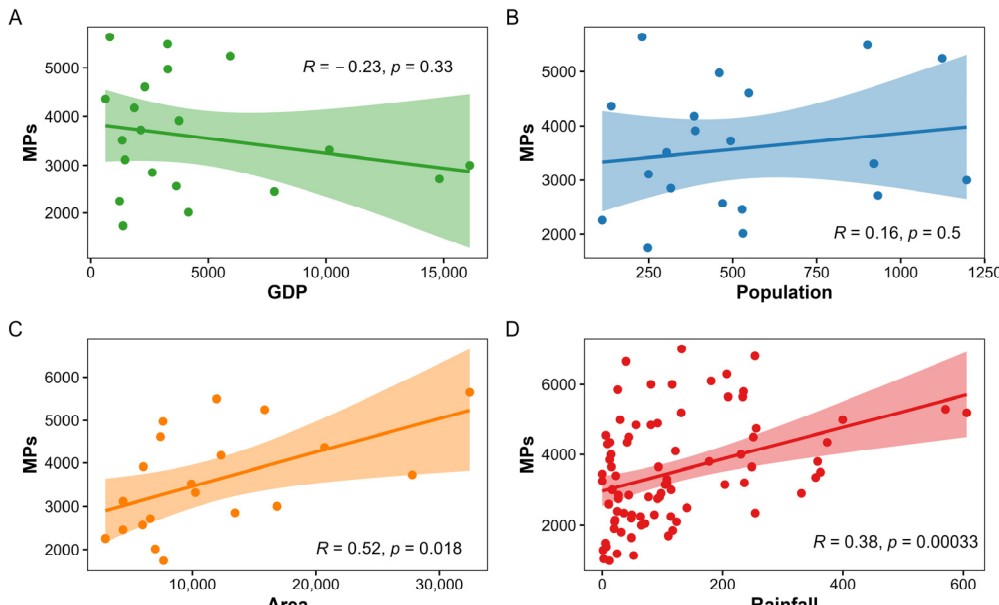

**Figure 10.** Regression analysis of microplastic abundance with social and natural factors. (**A**) Regression analysis of microplastic abundance with GDP. (**B**) Regression analysis of microplastic abundance with regional population. (**C**) Regression analysis of microplastic abundance with regional area. (**D**) Regression analysis of microplastic abundance with rainfall. (**A**–**C**) use the average values of four samplings. (**D**) uses all the values of four samplings, including microplastic abundance and rainfall.

## 4. Discussion

### 4.1. Spatial and Temporal Distribution Characteristics of Microplastic Pollution

This study discovered an overall trend of increasing microplastic abundance from west to east and from south to north in China's freshwater environments. Agricultural lands in China are mainly concentrated in the central, eastern, and northern regions [55], which is evident from the land type distribution shown in Figure 1. There is an inevitable correlation between the two. When categorizing all sampling points based on microplastic abundance from low to high, significant differences were observed between the groups ($p < 0.001$), with Group C sampling points showing the highest microplastic abundance, which also had the highest proportion of agricultural land. This precisely indicates that the possible source of microplastics in freshwater environments is the adjacent farmlands [56,57].

Overall, the abundance of microplastics increased first and then decreased from April to October, with the highest abundance recorded in June. This pattern corresponds with China's rainy season, which mainly occurs from June to August (summer). Although Group C did not experience the highest rainfall, the accumulation of microplastics during the dry season (spring) is expected to be higher than in the other groups.

In the dry season, microplastics accumulate on the soil surface or on roads covered with dust [58,59]. Kang et al. conducted a study in Goyang city, South Korea, and found that microplastic concentrations in road dust increased with the drying period, suggesting a significant accumulation of these pollutants on road surfaces in dry conditions [60]. Their research indicated that after a three-day drying period, the concentration of microplastics was notably higher, with a significant portion originating from vehicle tires and road materials.

During the rainy season, these microplastics are transported to freshwater environments with rainwater runoff [61–63]. Koutnik et al. conducted a global analysis on the distribution of microplastics in soil and freshwater environments, focusing on the factors affecting their concentration and the fundamental transport processes. Their findings indicate that microplastic concentrations in inland locations such as glaciers and urban stormwater were significantly higher than in rivers, suggesting the importance of rainwater runoff in microplastic transport [64].

Furthermore, regression analysis between regional population, area size, and microplastic abundance revealed a positive correlation, though not significant. However, there was no positive correlation between regional GDP and microplastic abundance, possibly because economically developed areas have a lower proportion of agricultural land and pay more attention to environmental protection, thereby limiting the use of plastic products [65–68].

### 4.2. Potential Sources of Microplastic Pollution

Both Mantel tests and redundancy analysis (RDA) indicate that the distribution characteristics of microplastics are directly related to land use types, predominantly concentrated in areas of frequent human activity, including agricultural, transport, and urban land.

Agricultural activities involve extensive use of plastic greenhouses and mulch films, mostly made of polyethylene (PE) or polyvinyl chloride (PVC) [69–71]. Zhang et al. found the agricultural plastic film usage in China in 2017 was 2,528,600 tons [72]. After agricultural film recycling and water erosion, the plastic debris amount was estimated as 465,016 tons. The water erosion process carried 4329 tons of plastic debris into the aquatic environment. Studies have shown that under the combined effects of sunlight and rainfall, these plastic films continuously break down into smaller pieces, forming microplastics that can migrate into deeper soil layers [73]. Additionally, the long-term application of sludge and organic fertilizers is another significant source of microplastic pollution in farmlands [74]. About 90% of microplastics in wastewater accumulate in sludge, which is often used as fertilizer after pretreatment [75]. However, conventional sludge pretreatment methods, such as anaerobic fermentation and heat drying, are ineffective in removing microplastics [76,77]. Thus, microplastics enter and accumulate in the soil through sludge used as fertilizer [78]. This explains why polyethylene and polyvinyl chloride are the main chemical components of microplastic samples in this study (Figure 4). Among all sampling points, Group C had the highest proportion of film-shaped microplastics and the largest average size, likely related to the high proportion of agricultural land in this group.

In transportation activities, microplastics primarily originate from two sources. The first source is road dust [79]. Su et al. found that the average abundance of microplastics in road dust collected from typical streets in Phillip Bay, Australia, and its upstream area during different precipitation seasons ranged from 20.6 to 529.3 items/kg [80]. Fibers (70.8%), individuals smaller than 1 mm (41.9%), and polymers like polyester and polypropylene (combined 26.3%) constituted the majority of microplastics. Monitoring road dust is an economical and effective method for preliminary screening of microplastic pollution levels from atmospheric or urban non-point source diffusion. Road dust has been proven to be an important site where microplastics enter the environment from non-point sources [60,79]. The second major source in transportation is tire and brake wear [81]. Evangeliou et al. conducted global simulations of the atmospheric transport of microplastic particles produced by road traffic, including tire wear particles (TWPs) and brake wear particles (BWPs) [82]. Their findings reveal a high transport efficiency of these particles to remote regions, suggesting a significant environmental impact far from their urban source areas. Recent research by Griffith University in Australia analyzed the quantity and type of tire wear particles (TWPs) in urban stormwater runoff [83]. As tires wear, they release particles of varying sizes, from visible rubber chunks to microplastics. Annually, 6.6 million tons of TWPs are released worldwide, becoming a significant source of microplastic pollution.

Urban areas, as major human settlements, continuously generate a vast amount of microplastic pollution [59]. This experiment showed that fibers are the predominant shape of microplastics in China's freshwater environments (42.5%). Similar findings have been reported in many studies, especially those investigating urban water bodies [84,85]. For instance, studies were conducted in the Ottawa River basin in Canada and the Rhine River basin in Germany, where fiber detection rates exceeded 60%, with some areas reaching up to 100% [86,87]. Additionally, Hu et al. investigated microplastic distribution in 25 small water bodies in the Yangtze River Delta urban agglomeration, including Shanghai and Zhe-

jiang [48]. The results showed that microplastics were universally present, with an average abundance of 0.5–21.5 items/liter, predominantly in fiber form, accounting for 87.8%. A significant source of high fiber content in freshwater environments is domestic scattered discharge, such as the washing of synthetic clothing [88,89]. Another significant source is sewage treatment plants, where a large amount of non-removed microplastics remaining in the effluent or sewage sludge enters the water and soil through direct discharge or sludge reuse [90]. Currently, most sewage treatment plants (STPs) have varying processes for removing fibrous microplastics and have not yet established a unified standard for effectively treating microplastics, which is a significant challenge in addressing urban microplastic pollution [91,92].

### 4.3. The Driving Role of Rainfall on the Distribution of Microplastics

In this study, regression analysis between the abundance of microplastics at all sampling points and the rainfall in different months shows a very strong correlation ($p < 0.001$). This indicates a positive correlation between the abundance of microplastics and the amount of rainfall. Moreover, the detection results at all sampling points vary with the seasons, showing significant differences in the abundance of microplastics across different months. Combined with the multi-variate statistical analysis of human activities and meteorological data, it was found that urbanization and precipitation significantly affect the abundance and distribution of microplastics. It can be assumed that there is a pattern where microplastics, originating from human activities including agricultural production, transportation, and daily life, first accumulate in the soil environment during the dry season, and then are washed away and transported by runoff during the rainy season, eventually entering freshwater environments.

Freshwater microplastics primarily come from land-based sources, which can be divided into point sources and non-point sources [93]. Point sources include sewage treatment plants, plastic manufacturing companies, etc., while non-point sources include farmlands, roads, residential buildings, and commercial areas [94]. The types and colors of microplastics in water environments change with different land pollution sources [40,95]. In the investigation of microplastics in Italy's Ofanto River, due to the impact of agricultural activities in the watershed, especially the use of plastic film, black film-shaped microplastics dominated [96]. This indirectly proves the driving role of rainfall on the distribution of microplastics. In this study, the color distribution ratio of the three groups of samples was relatively balanced, with no significant differences ($p > 0.05$). This may be due to different pollution sources near different sampling points, showing no clear pattern.

Apart from rainfall, wind is also an important driver affecting the distribution of microplastics [97]. This is because the density of microplastic particles is much lower than that of soil minerals like quartz, and they are less "sticky", making them less likely to be captured by moisture like soil minerals [98]. Therefore, microplastics are more easily carried away by the wind [99]. Sometimes, the wind may not be strong enough to lift dust, but it can still carry microplastics into the air [100]. A study showed that smaller plastic particles can travel farther in the atmosphere [101]. Microplastic particles of 10 μm or smaller tend to fall closer to their source, but many particles of 2.5 μm or smaller can be carried far from the source. We acknowledge recent findings by Xiao et al., which highlight the significant role of microplastic fiber shapes in their long-distance atmospheric transport [102]. These insights are particularly relevant to our discussion on the pathways and mechanisms of microplastic migration. In agricultural activities, wind can promote the degradation of plastic film into microplastics and facilitate the spread of microplastics in the atmosphere [103]. In transportation activities, two major sources of microplastics—road dust and tire wear—are also affected by wind in addition to being washed away by rainfall [104].

### 5. Conclusions

This study elucidated the spatial and temporal distribution characteristics and potential sources of microplastic pollution in China's freshwater environments. It was found that the abundance of microplastics generally increases from west to east and from south to north, with higher abundance observed during the rainy season (summer) compared to the dry season (spring and autumn). The main reason is that the distribution of microplastics is directly related to land use types, primarily originating from agricultural, transport, and urban land. The change in microplastics' abundance with the seasons is mainly driven by rainfall. However, the threshold of rainfall that triggers the migration of microplastics remains unclear, which is a direction for future research [105].

This study is the first to investigate the spatial and temporal distribution characteristics of microplastic pollution in China's freshwater environments on a national scale, enriching the data on microplastic pollution in China's freshwater environments and filling a research gap in this field. The findings of this study provide a solid scientific basis for the control and legislation of microplastics, thereby establishing reliable monitoring schemes and formulating effective measures to protect freshwater environments.

**Supplementary Materials:** The following supporting information can be downloaded at: https://www.mdpi.com/article/10.3390/w16091270/s1, Figure S1: Work flow; Table S1: Detailed information of sampling sites; Table S2: Land use types of sampling sites (%); Table S3: Group information and microplastic abundance (n/m$^3$).

**Author Contributions:** Conceptualization, supervision and funding acquisition, X.T.; software—data analysis and visualization, H.H.; writing—original draft preparation, H.H.; writing—review and editing, X.T.; investigation, H.H., S.C. (Sulin Cai), S.C. (Siyuan Chen), Q.L., P.W., R.Y., X.Z., B.Y., Y.J., T.C., Y.L., H.J., R.L., Q.C., Y.F., L.P., Y.C., W.H., Y.P., G.P. and J.Z. All authors have read and agreed to the published version of the manuscript.

**Funding:** This research was funded by National Key Research and Development Program of China, grant number 2023YFC3905604; National Natural Science Foundation of China, grant number 31870598 and 31530007.

**Data Availability Statement:** Data set available on request from the authors.

**Acknowledgments:** The authors would like to extend sincere gratitude to everyone who provided assistance during the sampling process. Special thanks are due to the authors' families for invaluable contributions, without which this work would not have been possible.

**Conflicts of Interest:** The authors declare no conflicts of interest.

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
