# Peer review of "Spatial and Temporal Distribution Characteristics and Potential Sources of Microplastic Pollution in China’s Freshwater Environments"

_water, doi:10.3390/w16091270_

Round 1
Reviewer 1 Report
Comments and Suggestions for Authors
This manuscript investigated the characteristics of microplastic pollution in freshwater environments of 21 major cities across China. The work is engaging and flows smoothly across all sections. I believe it could be accepted for publication in Water after minor revision.
I’ve listed my suggestions below.
Figure 1.
The small image on the right side (down): what do it represent?
I suggest expanding the caption: are black dots the sampling station? add in the caption.
Material and methods
Sampling methods: In this study surface water samples have been samples by bottles (1L). Is this a valid method? Add references to support the used method.
How did you calculate the regional area? add unit measure
Quality assurance
This paragraph is well written. I suggest rewriting just the last sentence: “Additionally, blank controls were set up to detect system and experimental errors”. How many blank controls? please indicate the number and if any contamination was detected.
Results
I suggest adding the number of suspected particles have been identified in each sampling station.
Lines 229 -231. Add results related to non-plastic items: cellulose, dyes detected etc.
Line 277: “When comparing the microplastic abundances across the four sampling instances, the variations were found to be highly significant (p < 0.05)”. I suggest deleting “highly” or replace “0.05” with 0.01
Figure 4: please rewrite the caption.
Figure 5. I suggest adding in the map the measure unit of abundance values.
Figures 6, 7. Y ax: please add the measure unit.
Reviewer 2 Report
Comments and Suggestions for Authors
This study provides an insightful exploration of microplastics pollution in freshwater environments across China, a topic of significant interest.
For line 29, kindly include references to substantiate the '29%' mentioned, enhancing the credibility of the presented data.
On line 47, I suggest verifying the term 'plastic greenhouse' to ensure its appropriate usage and context within the field.
Regarding line 66, the term "closer association" may need clarification. Are you referring to proximity in a physical sense rather than ecological or trophic levels, particularly in relation to human health and seafood?
At line 68, the phrase "little attention" may not fully capture the extent of research on freshwater microplastics. While it may be relatively less than oceanic studies, acknowledging the existing body of freshwater research could provide a balanced perspective.
On line 77, please revise "fragmented" to terms like "scattered" or "distributed" when referring to data presentation, for precise scientific communication.
For line 93, consider specifying if 'city' encompasses both urban and suburban areas. The term 'sites' might offer clearer delineation.
Lines 102-103 raise a question about the exclusion of winter from the study. A rationale for this seasonal selection would elucidate the study's scope and methodology.
Line 131 suggests a need for title correction; it appears the content does not reflect observational data as implied. Please revise for accuracy.
Lines 153 and 187: it would be beneficial to direct these links to the reference list, ensuring a coherent and navigable manuscript.
For Figure 2, enhancing the image resolution could improve its interpretability and impact.
In Table 2, to avoid confusion with the numerous percentages, a clearer delineation of what each percentage represents would aid comprehension.
Lines 256-257 and Figure 6: elucidating the scientific significance of this analysis, particularly regarding proximity to pollution sources, would contextualize the findings.
For Figure 7, exploring the possibility of comparing the same locations across different seasons could offer deeper insights into temporal variations.
On line 300, a brief explanation of the Mantel test would benefit readers unfamiliar with this statistical method.
Lines 304 and Figure 8 need clarification on whether the term should be 'transport' or 'transportation,' and what 'transport land' denotes. Aligning terminology with the illustrated content will enhance clarity.
The rationale for displaying seasons in Figure 8 warrants explanation to understand their relevance to the study's outcomes.
In Figure 9, clarifying the axes labels, RDA1 and RDA2, would provide clearer guidance on the data's interpretation.
For Figure 10, elucidating the methodology for determining rainfall's impact and addressing the scattered data can strengthen the analysis presented (the data somehow very scattered and hard to reach conclusion).
Finally, line 491-492: including recent significant publications like Xiao, S., Cui, Y., Brahney, J., Mahowald, N.M. and Li, Q., 2023. Long-distance atmospheric transport of microplastic fibres influenced by their shapes. Nature Geoscience, 16(10), pp.863-870., would underscore the study's engagement with current research.
These suggestions aim to enhance the manuscript's scientific rigor and reader comprehension through clear, precise, and contextually grounded modifications.
